# SEED-GRPO: SEMANTIC ENTROPY ENHANCED GRPO FOR UNCERTAINTY-AWARE POLICY OPTIMIZATION

## ABSTRACT

Group Relative Policy Optimization (GRPO) introduces a new paradigm for reinforcement learning in Large Language Models (LLMs), modifying PPO by eliminating the value model for efficient post-training. However, vanilla GRPO assigns equal weight to all prompts during policy updates, ignoring that supervision whose target answers are inconsistent with the model's existing parameter knowledge can increase hallucinations and degrade downstream performance. To address this limitation, we propose SEED-GRPO (**S**emantic **E**ntropy **E**nhance**D** GRPO), which explicitly measures LLMs' uncertainty and uses it to modulate the learning process. This enables conservative updates for high-uncertainty prompts (*e.g.*, beyond model knowledge) while preserving relatively higher signals for confident ones. Experimental results on five mathematical reasoning benchmarks (AIME24 **56.7**, AMC **68.7**, MATH **83.4**, Minerva **34.2**, and OlympiadBench **48.0**) and on four few-shot fine-grained image classification datasets demonstrate that SEED-GRPO achieves new state-of-the-art performance in average accuracy. The code, implementation details will be publicly released.

## 1 INTRODUCTION

Reinforcement learning (RL) emerges as a critical tool for fine-tuning Large Language Models (LLMs) [43, 18, 33, 1, 46, 66, 30, 52, 66, 62, 21, 54] to improve reasoning and accuracy on complex tasks. Leading LLMs such as OpenAI's GPT-4o and o1 [38], Google's Gemini [45], Anthropic's Claude 3 Opus [2], Qwen series [3, 10, 56, 55, 57, 74], and DeepSeek [12, 43, 18] all rely on RL techniques to enhance their capabilities beyond what is possible with supervised learning alone. These models demonstrate remarkable proficiency in domains requiring sophisticated reasoning, with RL serving as the key mechanism. Recent advances like Group Relative Policy Optimization (GRPO) [43, 18] achieve strong performance by leveraging multiple sampled answers per prompt to compute relative advantages within each group, eliminating the need for separate value models.

Despite recent progress, GRPO [43, 18] and its variants [33, 30, 60, 9, 65, 69] **assign equal importance to all training prompts** during optimization. Recent studies reveal that learning from prompts inconsistent with the LLMs' pre-existing knowledge can lead to negative effects. For instance, Ren *et al*. [40] demonstrate that instruction fine-tuning often fails when the supervision introduces knowledge inconsistent with the model's internal learned knowledge, and that injecting new knowledge may even degrade downstream performance. Similarly, Gekhman *et al*. [17] show that fine-tuning on prompts containing factual knowledge unknown to the model not only converges significantly slower, but also increases the tendency of hallucination once learned. These findings suggest that forcing equal learning signals on all training prompts can degrade overall performance. In other words, during optimization, we should reduce the learning intensity for samples that are inconsistent with parametric knowledge and allow LLMs to focus more on learning those problems where they demonstrate consistent understanding.

This raises a critical question: how to identify which prompts are inconsistent with the model's existing parameter knowledge during training? We argue that LLMs' uncertainty [26, 16, 36, 14, 68, 36, 14] serves as a natural indicator. When a model generates semantically diverse and inconsistent responses to a prompt, it signals that the problem likely lies beyond the model's current knowl-

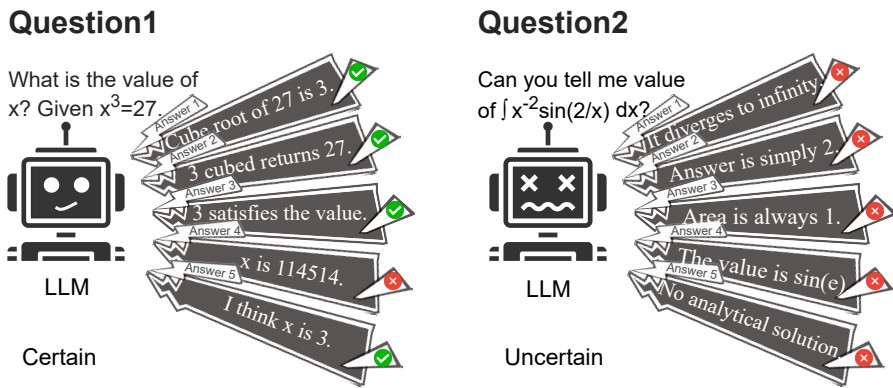

Figure 1: Intuitive explanation of semantic entropy. For Question 1, although the 5 responses have slight syntactic variations, 4 of them convey the same meaning, indicating low semantic entropy and high model certainty. For Question 2, the 5 responses can be clustered into 5 distinct meaning classes, resulting in high semantic entropy and indicating significant model uncertainty.

edge boundary—precisely the scenario that leads to negative learning dynamics identified in prior work [40, 17, 66, 16, 49, 58, 26, 24, 5]. Conversely, when a model produces semantically consistent responses, it indicates confident understanding, making such prompts suitable for intensive learning.

In this paper, we introduce SEED-GRPO (Semantic Entropy EnhanceD GRPO), an uncertainty-aware policy optimization algorithm that quantifies LLMs' epistemic uncertainty via semantic entropy [26, 16]. SEED-GRPO adaptively modulates policy updates at the prompt level: it applies conservative updates for high-uncertainty prompts while preserving relatively stronger learning signals for confident ones. This mechanism functions as a dynamic learning rate aligned with the model's knowledge boundary, similar to curriculum learning [4]. Semantic entropy integrates naturally with GRPO's sampling mechanism. Since GRPO already samples multiple responses per prompt to estimate relative advantages, semantic entropy exploits these same samples to quantify response diversity, which is almost a free lunch for GRPO. Moreover, increasing samples per prompt creates a synergistic effect—improving both advantage estimation and uncertainty quantification simultaneously (the ablation study can be found in Table 4(d)).

Our contributions are threefold: **i)** We identify semantic entropy as an effective indicator of knowledge consistency in LLMs: low entropy correlates with prompts aligned with the model's parametric knowledge (yielding correct predictions), while high entropy signals prompts beyond the knowledge boundary (resulting in inconsistent and incorrect outputs). **ii)** We introduce SEED-GRPO, an uncertainty-aware policy optimization algorithm that dynamically modulates policy updates based on semantic entropy. **iii)** We conduct a comprehensive empirical study on five mathematical reasoning benchmarks (AIME24, AMC, MATH [20], Minerva [27], and OlympiadBench [22]) and several fine-grained image classification datasets (Flower102 [37], Pets37 [39], FGVC [35], Cars196 [25]), empirically validating the efficacy of integrating uncertainty estimation into the reinforcement learning. Furthermore, we release our code and training configuration to facilitate future research in uncertainty-aware reasoning.

## 2 RELATED WORK

**Reasoning LLMs.** The development of reasoning capabilities in LLMs has emerged as a critical research area, with significant advances achieved through both sophisticated prompting innovations and training methods.

In the domain of prompting, the seminal Chain-of-Thought (CoT) [50] method significantly improved performance on mathematical and logical tasks by prompting models to generate intermediate reasoning steps. Building on this foundation, subsequent research has explored more sophisticated reasoning structures. For instance, the Tree-of-Thoughts (ToT) [58] framework organizes the reasoning process into a tree, allowing the model to explore, evaluate, and backtrack among multiple

reasoning branches at each step. Meanwhile, Self-consistency CoT [49] enhances the robustness and accuracy of results by sampling multiple independent reasoning paths and taking a majority vote on the final answer. The core of these methods lies in optimizing the search strategy at inference time to better unlock the model's existing potential.

While prompting techniques have shown remarkable success, recent efforts have shifted toward developing specialized reasoning models through targeted training approaches. LIMO [59] represents a significant advancement in this direction, employing Supervised Fine-Tuning (SFT) on carefully curated reasoning datasets to create models that inherently generate higher-quality reasoning chains. The approach demonstrates that models can learn to reason more effectively when trained on high-quality exemplars of step-by-step problem solving. The field has also witnessed breakthrough developments in reinforcement learning approaches for reasoning. Open-Reasoner-Zero [21] applies Monte Carlo Tree Search principles to reasoning, allowing models to explore and evaluate reasoning paths more systematically. Similarly, KIMI K1.5 [46] incorporates reinforcement learning from human feedback specifically tailored for reasoning tasks, while ReST-MCTS* [64] combines rejection sampling with Monte Carlo methods to improve reasoning quality through iterative refinement.

**Group Relative Policy Optimization and Variants.** DeepSeek introduced Group Relative Policy Optimization (GRPO) [43, 18], a reinforcement learning algorithm tailored for training reasoning LLMs. As a variant of Proximal Policy Optimization (PPO) [41], GRPO's primary innovation is its elimination of a value model, which is notoriously difficult to train and computationally expensive. This design has demonstrated strong performance on reasoning benchmarks across mathematics, coding, and question answering.

Following GRPO, several variants have been developed to address specific limitations or enhance its efficiency. To improve data efficiency, SRPO [69] incorporates history resampling to retain high-value problem instances for later training stages. Similarly, DAPO [60] employs dynamic sampling to focus training on more informative trajectories by filtering out those that are entirely correct or incorrect. Another line of improvement targets inherent biases; Dr.GRPO [33] identifies a length bias in the original algorithm and proposes modifications to mitigate it. The community has also contributed Open-R1 [15], a fully open-source implementation of GRPO. Visual-RFT [34] applies GRPO training to visual tasks, achieving notable results in fine-grained image recognition and object detection. GRPO-CARE [7] extends GRPO to multimodal large language models for video understanding, incorporating rollout consistency as a bonus reward into the reward function, whereas we use uncertainty to modulate the advantage.

Several other works incorporate entropy and uncertainty estimation into the GRPO framework [47, 66, 8, 11, 72, 76, 60, 75, 48, 19]. TTRL [76], INTUITOR [72], and EMPO [66] leverage model confidence, enabling unsupervised RL fine-tuning without relying on labeled annotations. DAPO [60], Cheng *et al.* [8], Cui *et al.* [11], and Wang *et al.* [48] identify that entropy collapse during RL training leads to premature exploration termination, thereby limiting RL performance. These works focus on using Shannon entropy [42] to maintain exploration diversity.

In contrast, our method leverages semantic entropy as a proxy for uncertainty to modulate the advantage calculations during the policy update, dynamically adapting the training signal to the model's confidence levels without additional computational overhead.

## 3 SEED-GRPO: UNCERTAINTY-AWARE POLICY OPTIMIZATION

### 3.1 MOTIVATION: UNCERTAINTY-AWARE LEARNING

The fundamental insight behind our approach is that when a model generates divergent responses to the same prompt across multiple attempts, such variation often reflects high uncertainty, suggesting that the prompt potentially exceeds the model's current knowledge (§1). SEED-GRPO leverages this insight through a principled mechanism: For prompts where the model exhibits high semantic entropy (high uncertainty), we adaptively downscale the advantages during policy updates, resulting in more conservative learning steps. This prevents the model from overfitting to potentially noisy rewards on prompts it cannot yet reliably solve. For questions where the model demonstrates low semantic entropy (high certainty), we maintain the original GRPO advantages.

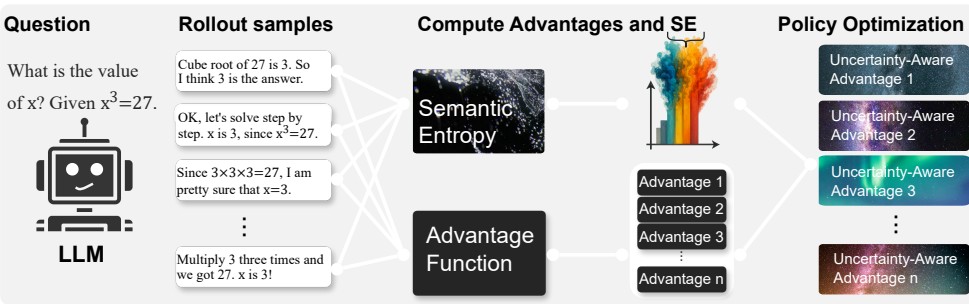

Figure 2: The SEED-GRPO framework incorporating semantic entropy for uncertainty-aware reinforcement learning. The framework samples multiple responses from a pre-trained LLM, computes semantic entropy to measure model uncertainty, and modulates the advantage function accordingly to enable more conservative updates for high-uncertainty questions.

This design echoes the principle of curriculum learning [4], where learning progresses from easier to harder examples. However, rather than relying on static difficulty heuristics, SEED-GRPO employs semantic entropy as a dynamic, model-specific uncertainty signal to calibrate learning pressure.

SEED-GRPO works in three steps: (1) sample multiple responses and compute GRPO advantages, (2) measure semantic entropy to quantify uncertainty of the prompt, (3) modulate advantages based on uncertainty before policy updates.

## 3.2 SEED-GRPO ILLUSTRATION VIA MATH REASONING EXAMPLE

To illustrate the core mechanics of SEED-GRPO, consider a math problem $q$ (prompt) such as:

"What is the value of $x$? Given $x^3 = 27$."

Using an LLM $\pi_{\theta_{\text{old}}}$, we sample a group of $N$ responses $\{o_1, o_2, \ldots, o_N\} \sim \pi_{\theta_{\text{old}}}(\cdot \mid q)$, as shown in Fig. 2, responses are also referred as rollout samples. Each response $o_i$ is a token sequence of length $l_i$, i.e., $o_i = (o_{i,1}, \ldots, o_{i,l_i})$. These sequences contain detailed step-by-step reasoning and conclude with a boxed final answer. While such sequences are sometimes referred to as "trajectories" in traditional reinforcement learning (e.g., PPO [41]), we avoid this terminology.

Each $o_i$ is an independently sampled text sequence. Some may contain correct solution paths, while others may contain logical or arithmetic errors. We extract the final answers and compute rewards: $r_i = 1$ if $o_i$ is correct, and $r_i = 0$ otherwise. Note that, in SEED-GRPO there is no reward model, these rewards are obtained by comparing with ground truth labels using specific verification rules. Following Dr. GRPO [33], we adopt a group-baseline advantage without standard-deviation normalization, where $\bar{r} = \frac{1}{N} \sum_{j=1}^{N} r_j$, and group relative advantages can be calculated:

$$A_i = r_i - \bar{r}, \quad A_i \in \mathbb{R}. \tag{1}$$

In SEED-GRPO, the advantage $A_i$ is broadcast across all tokens in the response $o_i$, i.e., each token $o_{i,t}$ in the same response shares the same scalar advantage $A_i$. For each token, we define the per-token importance ratio:

$$\text{ratio}_{i,t}(\theta) = \frac{\pi_\theta(o_{i,t} \mid q, o_{i,<t})}{\pi_{\theta_{\text{old}}}(o_{i,t} \mid q, o_{i,<t})}.$$

Following Dr.GRPO [33], the GRPO-style clipped surrogate objective used in SEED-GRPO is

$$\mathcal{L}(\theta) = \frac{1}{N} \sum_{i=1}^{N} \frac{1}{Z} \sum_{t=1}^{l_i} \min\Big(\text{ratio}_{i,t}(\theta)\, A_i, \ \text{clip}\big(r_{i,t}(\theta), 1-\epsilon, 1+\epsilon\big) A_i\Big), \tag{2}$$

where $Z$ is a constant normalization factor (e.g., the maximum generation length). Compared to the original GRPO formulation [43], Dr.GRPO and SEED-GRPO remove both the sequence-length and standard-deviation normalization terms, yielding an unbiased PPO-style objective with a group-baseline advantage.

To incorporate uncertainty into the learning process, we measure the **semantic entropy** $\text{SE}(q)$ [26, 16] of the generated answer group (rollout samples Fig. 2). Semantic entropy quantifies the degree of semantic diversity across the generated responses. It captures whether the outputs consistently converge on a single reasoning path or instead diverge into multiple, potentially conflicting solutions.

Intuitively, semantic entropy measures how diverse the model's sampled responses are in terms of their meaning (Fig. 1). Given a prompt $q$, we sample $N$ outputs $\{o_1, \dots, o_N\} \sim \pi_{\theta_{\text{old}}}(\cdot \mid q)$ and cluster them into $K$ semantic clusters $\mathcal{C} = \{C_1, \dots, C_K\}$, where each cluster groups responses that share the same meaning. For instance, in Fig. 1, the five responses to Question 1 collapse into two semantic clusters ($K = 2$), while the five responses to Question 2 form five distinct semantic clusters ($K = 5$).

The semantic entropy is theoretically defined as:

$$\text{SE}(q) = -\sum_{C_k \in \mathcal{C}} \Big( \sum_{o_i \in C_k} p(o_i \mid q) \Big) \log \Big( \sum_{o_i \in C_k} p(o_i \mid q) \Big), \tag{3}$$

where $p(o_i \mid q)$ is the probability of response $o_i$ given question $q$ under the policy model $\pi_{\theta_{\text{old}}}$.

In practice, we only observe a finite set of $K$ clusters. Following Kuhn et al. [26] and Farquhar et al. [16], we approximate the semantic entropy as the Shannon entropy of the induced cluster distribution:

$$\text{SE}(q) \approx -\sum_{k=1}^{K} \hat{P}(C_k \mid q) \, \log \hat{P}(C_k \mid q), \tag{4}$$

where $\hat{P}(C_k \mid q) = \frac{\sum_{o_i \in C_k} p(o_i \mid q)}{\sum_{j=1}^{K} \sum_{o_i \in C_j} p(o_i \mid q)}$ is the normalized probability mass of cluster $C_k$.

Semantic entropy is non-negative and measures the model's uncertainty on the given prompt. When all $N$ responses convey the same meaning ($K{=}1$), the entropy reaches its minimum value of 0, indicating complete certainty. Conversely, when each response belongs to a distinct semantic cluster ($K = N$), the entropy reaches its maximum value, signaling extreme uncertainty where the model produces entirely different answers each time. Given a fixed number of responses $N$, the theoretical upper bound for a fixed sample size $N$ is $\text{SE}_{\max} = \log N$, which occurs when all responses form distinct clusters ($K = N$) with uniform probability mass. For instance, with $N = 8$ and a uniform distribution across distinct clusters, the maximum semantic entropy is approximately **2.08**.

This semantic entropy allows us to quantify the uncertainty of the model for each prompt. Higher entropy indicates greater semantic diversity in the model's responses, suggesting that the model is uncertain about the given prompt. Lower entropy indicates greater consensus among responses, suggesting higher confidence in the model's answers. We leverage this uncertainty measurement to modulate the advantage in the reinforcement learning objective. The key insight is that model updates should be more conservative for questions where the model exhibits high uncertainty. Our uncertainty-aware advantage modulation function is defined as:

$$\hat{A}_i = A_i \cdot f\big(\alpha \cdot \text{SE}(q)/\text{SE}_{\max}\big), \tag{5}$$

where $\alpha$ is a hyperparameter controlling sensitivity. When semantic entropy is high, we interpret it as model uncertainty and scale down the advantage to produce more conservative updates. The function $f$ can take various forms (see Appendix D), such as linear, exponential, or focal styles, influencing how uncertainty affects the advantage scaling. We conduct ablation studies on $f$ in detail (§4.3).

Intuitively, this approach makes the training process more cautious about learning from prompts where the model lacks confidence, mitigating the risk of overfitting to potentially noisy supervision.

## 3.3 DISCUSSION AND ANALYSIS

**1) Information entropy or Semantic entropy?**

Shannon entropy [42] is widely used in reinforcement learning for LLM training [8, 11, 60, 75, 48, 19, 44, 53, 51, 70, 13, 29, 23, 71]. These studies report that policy entropy often collapses during optimization, leading to premature convergence and insufficient exploration. By explicitly incorporating Shannon entropy into the objective, they successfully encourage exploration and stabilize

training. Our work is orthogonal: rather than focusing on token-level entropy to promote exploration, we leverage *semantic entropy* to capture prompt-level uncertainty of LLMs. This semantic perspective allows us to identify potentially harmful prompts. Importantly, information entropy and semantic entropy address distinct challenges and are, in principle, complementary. Future work may integrate both dimensions to achieve exploration-aware and uncertainty-aware policy optimization.

**2) What benefits does uncertainty-aware advantage bring to policy optimization?**

Incorporating uncertainty into the advantage computation allows SEED-GRPO to modulate the learning process adaptively. To better understand this mechanism, we present a simplified gradient analysis of the policy update. For clarity, we consider the loss function without clipping:

$$\mathcal{L}_i(\theta) = \text{ratio}_i(\theta) \cdot \hat{A}_i = \frac{\pi_\theta(o_i \mid q)}{\pi_{\theta_{\text{old}}}(o_i \mid q)} \cdot \hat{A}_i. \tag{6}$$

The gradient is computed as:

$$\nabla_\theta \mathcal{L}_i(\theta) = \nabla_\theta \log \pi_\theta(o_i \mid q) \cdot \text{ratio}_i(\theta) \cdot \hat{A}_i. \tag{7}$$

Accordingly, the policy update becomes (with the global learning rate $\eta$):

$$\theta \leftarrow \theta + \eta \cdot \nabla_\theta \log \pi_\theta(o_i \mid q) \cdot \text{ratio}_i(\theta) \cdot \underbrace{\left[ A_i \cdot f\left( \alpha \cdot \text{SE}(q)/\text{SE}_{\max}(q) \right) \right]}_{\hat{A}_i}. \tag{8}$$

By integrating semantic uncertainty into $\hat{A}_i$, this formulation effectively scales the gradient for each input based on the model's uncertainty. This uncertainty-aware advantage computation effectively creates a **prompt-specific adaptive learning rate**.

As shown in Eq. 8, the policy update is governed by four components: the global learning rate $\eta$, the log-probability gradient, the importance sampling ratio, and the advantage term. By incorporating the uncertainty-dependent factor $f(\cdot)$, which is non-negative, SEED-GRPO effectively modulates the update magnitude in proportion to the model's uncertainty. This can be viewed as dynamically adjusting the effective learning rate on a per-prompt basis.

This mechanism creates an implicit curriculum learning effect: the model naturally takes larger learning steps on problems it can confidently solve, while proceeding more cautiously on challenging ones where the reward signal may be less reliable. This approach helps prevent overfitting to noise in difficult problems while allowing efficient learning from well-understood ones.

## 4 EXPERIMENTS

### 4.1 EXPERIMENTAL SETUP

**Train.** For mathematical reasoning, we use the Level 3–5 subset of the MATH benchmark [20], following the same setting as Dr.GRPO [33]. For fine-grained image classification, we follow GPG [9] and Visual-RFT [34] and evaluate on several standard datasets, including Flowers102 [37], Pets37 [39], FGVC [35], and Cars196 [25].

**Test.** We evaluate our method on five mathematical reasoning benchmarks: **i)** AIME24 contains 30 high-school level olympiad problems from the American Invitational

Table 1: Dataset statistics.

| Dataset | #Questions | Level |
|---|---|---|
| *Train Datasets* | | |
| MATH (L3–L5) | 8.5k | – |
| *Test Datasets* | | |
| AIME24 | 30 | Olympiad |
| AMC | 83 | Intermediate |
| MATH500 | 500 | Advanced |
| Minerva | 272 | Graduate |
| OlympiadBench | 675 | Olympiad |

Mathematics Examination 2024; **ii)** AMC includes 83 problems from the AMC series, consisting mostly of multiple-choice questions of intermediate difficulty; **iii)** MATH500 is a randomly selected subset of 500 problems from the original MATH [20] dataset, covering algebra, geometry, and number theory; **iv)** Minerva (MIN) [27] comprises 272 questions introduced by the Minerva benchmark mostly requiring multi-step reasoning; **v)** OlympiadBench (OLY) [22] includes 675 high-difficulty math problems.

**Model.** Following previous works [18, 33], we use Qwen2.5-Math [56] 1.5B, 7B, and DeepSeek-R1-Distill-Qwen-7B [18] as our base models. Dr.GRPO [33] is the default baseline algorithm.

Table 2: Pass@1 performance comparison across multiple mathematical reasoning benchmarks. Results marked with [+] are reported as the mean $\pm$ standard deviation across 5 runs under the same default experimental setting (§4.1). Our other results report the best performance.

| Method | AIME24 | AMC | MATH | MIN. | OLY. | Avg. |
|---|---|---|---|---|---|---|
| *Baseline methods* | | | | | | |
| Qwen2.5-Math-base **1.5B** 🐟 | 16.7 | 43.4 | 61.8 | 15.1 | 28.4 | 33.1 |
| Qwen2.5-Math-base **7B** 🐟 | 0.2 | 45.8 | 69.0 | 21.3 | 34.7 | 34.2 |
| GRPO w/ Entropy Adv. **7B** 🖼 | 33.7 | 69.8 | 83.1 | - | - | - |
| GRPO w/ KL-Cov **7B** 🖼 (Avg@32) | 22.6 | 61.4 | 80.8 | 38.2 | 42.6 | 49.1 |
| EMPO **7B** 🐧 | 20.0 | 65.0 | 78.0 | 40.4 | 37.3 | 48.1 |
| FR3E **7B** 📊 | 39.1 | 67.5 | 82.2 | 40.8 | 46.5 | 55.2 |
| Dr.GRPO **1.5B** 🔵 | 20.0 | 53.0 | 74.2 | 25.7 | 37.6 | 42.1 |
| Dr.GRPO **7B** 🔵 | 43.3 | 62.7 | 80.0 | 30.1 | 41.0 | 51.4 |
| RAFT++ **7B** 🗡 | - | - | 80.5 | 35.8 | 41.2 | - |
| OpenReasoner-Zero **7B** 🚀 | 13.3 | 47.0 | 79.2 | 31.6 | 44.0 | 43.0 |
| Eurus **7B** 🐿 | 16.7 | 62.7 | 83.8 | 36.0 | 40.9 | 48.0 |
| SimpleRL-Zoo **7B** 🐯 | 26.7 | 60.2 | 78.2 | 27.6 | 40.3 | 46.6 |
| GPG **7B** 📨 | 33.3 | 65.0 | 80.0 | 34.2 | 42.4 | 51.0 |
| SRPO **32B** 🌀 | 44.3 | - | - | - | - | - |
| DAPO **32B** 📊 (Avg@32) | 50.0 | - | - | - | - | - |
| DeepSeek-R1-Zero-Qwen **32B** 🐋 | 46.7 | - | - | - | - | - |
| QwQ-preview **32B** 🐟 | 50.0 | - | 90.6 | - | - | - |
| Beyond 80/20 **8B** 🐟 (Avg@16) | 34.58 | 77.19 | 89.70 | 40.26 | 57.43 | 59.8 |
| GMPO **7B** 🖼 | 43.3 | 61.4 | 82.0 | 33.5 | 43.6 | 52.7 |
| GRPO-LEAD **7B** 🏆 | 47.0 | 74.8 | 87.0 | 37.2 | 50.0 | 59.2 |
| DisCO **7B** (R1-Distill, 8k length) 🅰️ | 55.8 | 85.4 | 92.7 | 41.0 | 59.2 | 66.8 |
| *Our methods* 🎯 | | | | | | |
| SEED-GRPO **1.5B** (Linear, $\alpha$=0.02) | 23.3 | 50.6 | 75.4 | 26.8 | 41.3 | 43.5 |
| SEED-GRPO **7B** (Linear, $\alpha$=0.02)[+] | $43.3_{\pm3.4}$ | $64.67_{\pm4.9}$ | $82.2_{\pm1.4}$ | $35.03_{\pm1.6}$ | $45.2_{\pm2.2}$ | $54.73_{\pm2.0}$ |
| SEED-GRPO **7B** (Linear, $\alpha$=0.02) | 46.7 | 69.9 | 83.0 | 36.7 | 46.8 | 56.6 |
| SEED-GRPO **7B** (Linear, $\alpha$=0.02, $G$=16) | **56.7** | 68.7 | **83.4** | 34.2 | 48.0 | 58.2 |
| SEED-GRPO **7B** (Linear, $\alpha$=0.02, R1-Distill) | 50.0 | **78.3** | 91.6 | 38.6 | 61.5 | 64.0 |
| SEED-GRPO **7B** (Linear, $\alpha$=0.02, R1-Distill, 8k length) | **63.3** | 74.7 | **93.2** | **40.4** | **65.3** | **67.4** |

Table 3: Training configuration and performance comparison of mathematical reasoning methods.

| Method | #Train Data | #Prompt Batch Size | #Rollouts($G$) | #Steps | AIME24 | MATH |
|---|---|---|---|---|---|---|
| *Baseline methods* | | | | | | |
| Dr.GRPO **7B** 🔵 | 8.5k | 128 | 8 | 400 | 43.3 | 80.0 |
| SimpleRL-Zoo **7B** 🐯 | 7.5k | 1024 | 8 | 150 | 26.7 | 78.2 |
| DAPO **32B** 📊 | 17k | 512 | 16 | 5.5k | 50.0(Avg@32) | - |
| *Our methods* 🎯 | | | | | | |
| SEED-GRPO **7B** | 8.5k | 128 | 8 | 384 | 40.0 | 81.4 |
| SEED-GRPO **7B** | 8.5k | 128 | 8 | 928 | 46.7 | 83.0 |
| SEED-GRPO **7B** | 8.5k | 128 | 16 | 360 | **56.7** | **83.4** |
| SEED-GRPO **7B** (R1-Distill, 8k length)) | 8.5k | 128 | 8 | 1072 | **63.3** | **93.2** |

**Competitor.** We compare against state-of-the-art methods including Dr.GRPO [33], DeepSeek-R1-Zero-Qwen [43], RAFT++ [52], GPG [9], DAPO [60], SimpleRL-Zoo [63], SRPO [69], Eurus [61], OpenReasoner-Zero [21], and QwQ-preview [55], GRPO w/ Entropy [8], GRPO w/ KL-Cov [11], EMPO [66], FR3E [75], Beyond 80/20 [48], GMPO [73], GRPO-LEAD [65], and DisCO [28].

**Evaluation Metrics.** To maintain consistency with prior research [33, 63], we primarily employ the Pass@1 metric for comparative analysis [6]. The pass@$k$ metric evaluates whether, among $k$ responses to a given problem, at least one solution passes the test criteria. The Pass@1 scenario, where only a single response is generated, presents a more challenging setting. For the uncertainty function $f(\cdot)$, we default choose Linear function with $\alpha = 0.02$, more ablation studies are in §4.3.

**Implementation Details.** Our training configuration follows Dr.GRPO [33]. Specifically, we limit the maximum output to 3,000 tokens, and when calculating advantages, we do not normalize by the group reward standard deviation. Similarly, during loss computation, we do not divide by generation length. For semantic entropy clustering, we employ a straightforward approach that only considers whether the final answers generated by the model are identical (Appendix A). All experiments are conducted on a server equipped with 8 NVIDIA A800 GPUs (80GB each) (Appendix C).

## 4.2 QUANTITATIVE COMPARISON RESULTS

Table 2 presents a comprehensive evaluation of our SEED-GRPO approach against established mathematical reasoning methods across multiple benchmarks. Our method demonstrates consistent and substantial improvements over strong baseline systems. Under the Qwen-Math-base setting,

SEED-GRPO 1.5B shows significant average improvements compared to the Qwen-Math-base1.5B model, achieving 43.5% average score across all benchmarks.

For our default configuration (§4.1), SEED-GRPO 7B (Linear, $\alpha$=0.02) achieves an excellent average score of 56.6% across all benchmarks, representing a significant improvement of **5.2%** over the Dr.GRPO 7B baseline. Notably, SEED-GRPO 7B even surpasses SRPO 32B on the challenging AIME24 benchmark (46.7% vs. 44.3%), despite having only a fraction of the parameters. This configuration particularly excels on the AMC benchmark with a score of 69.9%, surpassing all other 7B parameter models with the same initial base architecture.

Our experiments further validate the effectiveness of increasing the number of rollouts $G$ per query. As shown in Table 2, simply doubling $G$ from 8 to 16 leads to a **+1.6**% gain on average score, and a dramatic **+10**% jump on AIME24 (from 46.7% to 56.7%). This enhanced configuration achieves an average score of 58.2% across all benchmarks, outperforming several 32B models including SRPO, DAPO, DeepSeek-R1-Zero-Qwen, and QwQ-preview. Importantly, these results come at a significantly lower computational cost compared to training large 32B models.

Notably, in the DeepSeek-R1-Distill-Qwen-7B setting, our SEED-GRPO (7B, R1-Distill) achieves the best overall performance, with an impressive average score of 64.0% on Pass@1. It outperforms all 7B and even 32B models across key benchmarks like AIME24, MATH, and OlympiadBench.

Table 3 compares performance across different training configurations. Compared to baseline methods, our SEED-GRPO achieves superior results with similar or even reduced training data size and computational steps. In particular, with $8.5k$ training data and a batch size of 128, by increasing the number of rollouts to 16, the AIME24 score improved to 56.7% and the MATH score reached 83.4%, surpassing all other 7B models.

It is worth highlighting that our SEED-GRPO 7B (Linear, $\alpha$=0.02) achieves superior performance to several 32B models AIME24, demonstrating the effectiveness of our approach. While DAPO reports a higher Avg@32 score of 50.0%, our method focuses on the more challenging Pass@1 metric.

## 4.3 ABLATION STUDY

**Method Comparison.** Table 4(a) compares SEED-GRPO with the initial base model Qwen2.5-Math-base 7B and the baseline Dr.GRPO 7B. It's important to note that both SEED-GRPO and Dr.GRPO start from the same Qwen2.5-Math-base 7B, using identical hyperparameters. Particularly, SEED-GRPO achieves a remarkable 13.4% improvement over the baseline on AIME (from 43.3% to 46.7%). On average, SEED-GRPO outperforms Dr.GRPO by 5.2% confirming the effectiveness of uncertainty-aware policy optimization.

**Semantic Entropy Weight.** We investigate the impact of the semantic entropy weight parameter $\alpha$ in Table 4(b), which controls how much influence uncertainty has on the training process. Our results indicate that a medium weight value of $\alpha = 0.02$ yields the best overall performance with an average accuracy of 56.6%. Interestingly, a higher weight ($\alpha = 0.03$) improves performance on the challenging AIME benchmark but slightly decreases performance on other tasks. This suggests that more difficult tasks may benefit from stronger uncertainty weighting, while simpler tasks require less emphasis on uncertainty. Setting $\alpha$ too low (0.01) consistently underperforms, confirming that some degree of uncertainty modeling is beneficial across all benchmarks.

**Weight Function.** In Table 4(c), we evaluate different functional forms for incorporating semantic entropy into our training objective. We compare linear, exponential, and focal weighting functions (Appendix D). The linear weighting function achieves the best overall performance with an average accuracy of 56.6%, outperforming both alternatives. While the focal function excels on particular benchmarks like MATH (84.4%) and OLY (47.6%), it performs less consistently across all tasks. The exponential function shows competitive but generally lower performance, suggesting that more aggressive uncertainty penalization may not be optimal. These results indicate that a simple linear relationship between semantic entropy and policy updates provides the most robust learning signal.

**Number of Rollouts.** Table 4(d) examines how the number of sampled solutions per query ($G$) affects model performance. Increasing $G$ from 8 to 16 improves the average accuracy from 56.6% to 58.2%, with particularly gains on the challenging AIME benchmark (from 46.7% to 56.7%). This improvement demonstrates that a larger sample size enables more accurate estimation of se-

**(a) Method Comparison**

| Method | AIME | AMC | MATH | MIN | OLY | Avg. |
|---|---|---|---|---|---|---|
| Baseline **7B** | 0.2 | 45.8 | 69.0 | 21.3 | 34.7 | 38.2 |
| Dr.GRPO **7B** | 43.3 | 62.7 | 80.0 | 30.1 | 41.0 | 51.4 |
| SEED-GRPO | 46.7 | 69.9 | 83.0 | 36.7 | 46.8 | 56.6 |

**(b) SE Weight $\alpha$**

| $\alpha$ | AIME | AMC | MATH | MIN | OLY | Avg. |
|---|---|---|---|---|---|---|
| 0.01 | 46.7 | 60.2 | 80.6 | 33.5 | 42.7 | 52.7 |
| 0.02 | 46.7 | **69.9** | **83.0** | **36.7** | **46.8** | **56.6** |
| 0.03 | **50.0** | 61.4 | 83.0 | 34.2 | 44.4 | 54.6 |

**(c) Weight Function $f(\cdot)$**

| Func. | AIME | AMC | MATH | MIN | OLY | Avg. |
|---|---|---|---|---|---|---|
| Focal | 43.3 | 65.1 | **84.4** | 35.3 | **47.6** | 55.1 |
| Exponential | 43.3 | 66.3 | 82.0 | 35.7 | 44.3 | 54.3 |
| Linear | 46.7 | **69.9** | 83.0 | **36.7** | 46.8 | **56.6** |

**(d) #Rollouts ($G$)**

| $G$ | AIME | AMC | MATH | MIN | OLY | Avg. |
|---|---|---|---|---|---|---|
| 8 | 46.7 | **69.9** | 83.0 | 36.7 | 46.8 | 56.6 |
| 10 | 50.0 | 61.4 | **84.0** | **37.5** | **48.1** | 56.2 |
| 16 | **56.7** | 68.7 | 83.4 | 34.2 | 48.0 | **58.2** |

**(e) Base Models**

| Method | AIME | AMC | MATH | MIN | OLY | Avg. |
|---|---|---|---|---|---|---|
| *Qwen2.5 1.5B* | | | | | | |
| Base | 16.7 | 43.4 | 61.8 | 15.1 | 28.4 | 33.1 |
| Dr.GRPO **1.5B** | 20.0 | 53.0 | 74.2 | 25.7 | 37.6 | 42.1 |
| SEED-GRPO | 23.3 | 50.6 | 75.4 | 26.8 | 41.3 | 43.5 |
| *Qwen2.5 7B* | | | | | | |
| Base | 0.2 | 45.8 | 69.0 | 21.3 | 34.7 | 38.2 |
| Dr.GRPO | 43.3 | 62.7 | 80.0 | 30.1 | 41.0 | 51.4 |
| SEED-GRPO | **56.7** | 68.7 | 83.4 | 34.2 | 48.0 | 58.2 |
| *R1-Distill 7B* | | | | | | |
| Base | 10.0 | 26.2 | 80.0 | 30.1 | 41.0 | 51.4 |
| SEED-GRPO | 50.0 | **78.3** | **91.6** | **38.6** | **61.5** | **64.0** |

**(f) Fine-grained Image Classification**

| Models | Avg. | Flower102 | Pets37 | FGVC | Cars196 |
|---|---|---|---|---|---|
| Qwen2-VL-2B | 56.0 | 54.8 | 66.4 | 45.9 | 56.8 |
| + SFT | 55.6 | 58.5 | 55.5 | 67.9 | 40.5 |
| + GRPO | 81.9 | 71.4 | 86.1 | 74.8 | 95.3 |
| + GPG | 86.0 | 73.0 | 87.1 | 86.8 | 97.1 |
| + SEED-GRPO | **88.5** | **78.2** | **89.3** | **88.9** | **97.7** |

**(g) Training Dynamics of Reverse SEED-GRPO**

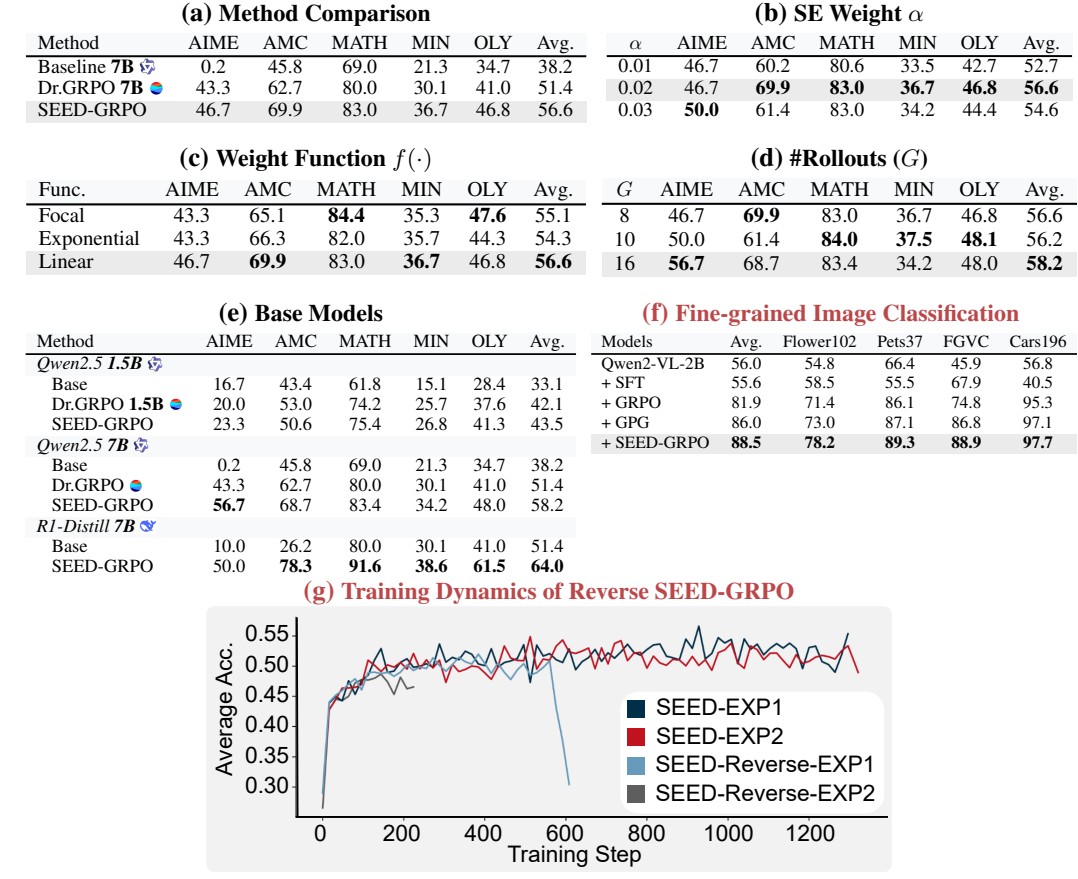

Table 4: SEED-GRPO ablations across five math reasoning benchmarks and fine-grained image classification tasks.

mantic entropy. However, the performance with $G = 10$ shows mixed results, performing best on some benchmarks (MATH, MIN, OLY) but worse on others (AMC), suggesting task-specific optimal sampling strategies. Overall, our findings support using larger rollout numbers when computational resources permit, with diminishing returns likely beyond $G = 16$.

**Base Models.** Table 4(e) shows SEED-GRPO's effectiveness across different base models. When applied to Qwen2.5-1.5B, SEED-GRPO improves average performance by 10.4 percentage points (from 33.1% to 43.5%). The improvement is even more substantial for Qwen2.5-7B, with a 20.0 percentage point gain (from 38.2% to 58.2%). This demonstrates that SEED-GRPO's benefits scale with model size, suggesting that larger models can better leverage uncertainty information during training. We also evaluated SEED-GRPO on the R1-Distill 7B model, achieving strong performance on AMC (78.3%) and AIME (50.0%).

**4-shot Fine-grained Image Classification.** Table 4(f) reports 4-shot results on fine-grained image classification with Qwen2-VL-2B. SEED-GRPO achieves the best overall performance, attaining an average of 88.5% and consistently outperforming GPG on all four datasets, with the largest gain on Flowers102 (78.2% vs. 73.0%). These results indicate that semantic-entropy–guided updates transfer effectively to the multimodal setting.

**Impact of Uncertain-Aware Advantage Modulation Strategy.** As shown in Table 4 Figure(g), in SEED-EXP1 and SEED-EXP2, we use the normal SEED-GRPO training, where we reduce the advantage for prompts with higher uncertainty. In the control groups SEED-Reverse-EXP1 and SEED-Reverse-EXP2, we employ the opposite strategy: if the model is more certain about a prompt, we reduce its advantage, while for prompts with higher uncertainty, we preserve relatively higher advantages. The y-axis represents the average accuracy across 5 mathematical benchmarks, and the

x-axis represents the training steps. The reverse version consistently performs worse than SEED throughout, and experiences model collapse at step 600.

## 5 LIMITATION AND FUTURE WORK

**Limitation.** Our current implementation of SEED-GRPO focuses solely on utilizing the final answers for semantic clustering in mathematical reasoning tasks, without considering the intermediate reasoning steps. This design choice offers simplicity and proves effective for problems with unique, well-defined answers. However, for open-ended problems without unique answers, our current semantic entropy calculation may not adequately capture the diversity of valid reasoning paths.

**Future Work.** SEED-GRPO focuses on mathematical reasoning tasks, where the final answer can be explicitly verified. A promising direction for future work is to extend SEED-GRPO to other domains such as multimodal tasks (image-text [31, 68, 67], video understanding [7]), code generation, and open-ended textual question answering. These domains often require more nuanced semantic understanding and may benefit even more from uncertainty-aware policy optimization.

## 6 CONCLUSION

We introduce SEED-GRPO, an uncertainty-aware policy optimization algorithm that integrates semantic entropy into GRPO to adaptively scale updates based on model confidence. Our method applies conservative updates to high-uncertainty prompts while maintaining effective learning on confident predictions. Experiments across five mathematical reasoning benchmarks and four fine-grained image classification datasets demonstrate that SEED-GRPO achieves new state-of-the-art results. Ablation studies confirm the effectiveness of uncertainty-aware policy optimization.

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

APPENDIX

For a better understanding of the main paper, we provide additional details in this supplementary material, which is organized as follows:

- §A provides the pseudo code of SEED-GRPO.

- §B discusses our limitations and directions of future work.

- §C provides the details of reproduction.

- §D discusses the detail forms of $f(\cdot)$.

## A  PSEUDO CODE

The pseudo-code of SEED-GRPO is given in Algorithm A. The code shows the method we use to calculate the semantic entropy given a prompt.

---
**Algorithm 1** Semantic Entropy Computation: PyTorch-like Pseudo-code

```
# Input: prompts, responses (candidates), log_liks
# Output: semantic_entropies for each question
def compute_semantic_entropy(prompts, candidates, log_liks):
    semantic_entropies = []
    # iterate over each question
    for q_idx, (prompt, responses, log_lik) in enumerate(zip(prompts, candidates,
        log_liks)):
        # Step 1:extract boxed answers
        question_answers = []
        for j in range(num_samples):
            ans = extract_answer(responses[j])
            if ans is not None:
                question_answers.append(ans)
            else:
                question_answers.append("NO_ANSWER_FOUND")
        # Step 2:handle cases based on answer validity
        if all(ans == "NO_ANSWER_FOUND" for ans in question_answers):
            # Case 1:all invalid answers -> maximize entropy
            semantic_ids = assign_unique_ids(question_answers)
        else:
            # split into valid and invalid
            valid_answers, no_answers = split_answers(question_answers)
            if no_answers and valid_answers:
                # Case 2:partial valid, partial invalid
                valid_semantic_ids = cluster_by_semantics(valid_answers)
                semantic_ids = merge_with_no_answer(valid_semantic_ids, no_answers)
            elif no_answers and not valid_answers:
                # Case 2 (edge case):all invalid after filtering
                semantic_ids = assign_unique_ids(question_answers)
            else:
                # Case 3:all valid answers
                semantic_ids = cluster_by_semantics(question_answers)
        # Step 3:compute semantic entropy
        log_lik_per_semantic = logsumexp_by_id(semantic_ids, log_lik, agg="sum_normalized")
        pe = predictive_entropy_rao(log_lik_per_semantic)
        semantic_entropies.append(pe)
    return semantic_entropies
```
---

## B  LIMITATION AND FUTURE WORK

**Limitation.** While SEED-GRPO effectively leverages final answers for semantic clustering in mathematical reasoning tasks, this approach simplifies the learning process at the cost of ignoring certain complexities. Specifically:

- **Diversity of valid solutions.** For questions with multiple acceptable answers, relying solely on final outputs may overlook alternative reasoning paths, potentially underrepresenting uncertainty. For example, for open-ended problems without unique answers, our current semantic entropy calculation can not be implemented by using the final answers to cluster responses.

- **Intermediate reasoning signals.** Excluding intermediate steps prevents the model from capturing nuances of multi-step reasoning, which could affect performance on complex compositional problems. Current implementation works well for mathematical domains with clear correctness criteria; it may be insufficient for domains requiring nuanced evaluation of the reasoning process itself.

- **Transfer to other domains.** The current design is tailored for well-defined mathematical tasks and may not fully capture uncertainty in domains where evaluation criteria are less explicit or subjective.

- **Dependence on clustering methods.** Semantic entropy estimates hinge on the accuracy of clustering, which may be affected by limitations of the similarity metrics or external models used.

- **Resource considerations.** Processing many samples for semantic clustering and entropy computation can be computationally demanding, particularly for large datasets or when applied at inference time.

**Future Work.** There are multiple avenues to strengthen and extend the SEED-GRPO framework:

- **Incorporating reasoning trajectories.** Enhancing the semantic entropy computation to include intermediate reasoning steps may allow for finer-grained modeling of uncertainty and improve learning dynamics.

- **Broadening application domains.** Adapting SEED-GRPO to other types of reasoning, including multimodal tasks, program synthesis, and open-ended question answering, could demonstrate the framework's utility beyond mathematical problems.

- **Augmenting semantic clustering.** Integrating additional models—commercial LLMs or open-source encoders—could enrich the semantic grouping process, leading to more accurate entropy estimation.

- **Entropy-informed inference.** Using semantic entropy at test time to dynamically adjust generation strategies or enable fallback mechanisms could make the model more robust under uncertainty.

- **Efficiency improvements.** Future work could explore approximations, sampling strategies, or distributed computation to reduce the computational cost of semantic entropy estimation. Furthermore, using other efficient uncertainty estimators could be a promising way.

Overall, these directions aim to make SEED-GRPO more flexible, domain-general, and effective in capturing uncertainty across a variety of reasoning tasks.

## C    REPRODUCIBILITY

Our code and pre-trained models will be made publicly available. All models were trained on a single server equipped with eight A800 GPUs, using the OAT-LLM [32] reinforcement learning framework. We will release the full implementation details of our code.

## D    FUNCTIONAL FORMS

In this appendix, we detail the specific functional forms used to modulate advantage estimates based on semantic entropy. Let $SE(q)$ denote the semantic entropy for a given prompt $q$, and $N$ be the number of sampled responses. We first define the **normalized semantic entropy** $\tilde{s}(q) \in [0, 1]$ as:

$$\tilde{s}(q) = \frac{SE(q)}{SE_{\max}}, \quad \text{where} \quad SE_{\max} = \ln N \tag{9}$$

represents the theoretical maximum entropy (assuming a uniform distribution over $N$ distinct clusters). The weighting function $f(\tilde{s}(q))$ takes the following forms:

**1. Linear.** The linear form applies a direct penalty proportional to the uncertainty:

$$f_{\text{linear}}(\tilde{s}(q)) = 1 - \alpha \cdot \tilde{s}(q), \tag{10}$$

where $\alpha$ is a hyperparameter controlling the modulation strength, in our experiments, we typically set $\alpha \in \{0.01, 0.02, 0.03\}$.

**2. Exponential.** This form utilizes an exponential decay function to provide a smoother penalty:

$$f_{\text{exp}}(\tilde{s}(q)) = \exp\left(-\alpha \cdot \tilde{s}(q)\right), \tag{11}$$

**3. Focal.** Inspired by Focal Loss, this variant imposes a stronger penalty on high-uncertainty samples to aggressively suppress their contribution:

$$f_{\text{focal}}(\tilde{s}(q)) = (1 - \tilde{s}(q))^{\gamma}, \tag{12}$$

where $\gamma > 0$ is a focusing parameter. We set $\gamma = 2.0$ in all experiments. Unlike the linear and exponential forms, which maintain weights close to 1.0, the focal function significantly down-weights highly uncertain prompts (where $\tilde{s}(q) \to 1$).

