# OpenReview forum: "SEED-GRPO: Semantic Entropy Enhanced GRPO for Uncertainty-Aware Policy Optimization"
_ICLR.cc/2026/Conference — Submitted to ICLR 2026_

### Official Review · Reviewer_q1DN · 2025-10-28

**Soundness:** 2
**Presentation:** 3
**Contribution:** 3
**Rating:** 4
**Confidence:** 3

**Summary:**

This paper studies the importance of different prompt in GRPO training. The paper first argues that training on some prompts on which the model is uncertain harms the training procedure. To mitigate this problem, the paper propose to use semantic entropy as an additional importance factor to reweight each prompt. The experiments on three models shows that proposed method outperforms a variety of baselines.

**Strengths:**

The strengths of this paper is shown as follows

1. This paper propose SEED-GRPO, which mitigate the issue of training on potential harmful prompt in a simple way.

2. The experiments are conducted on three models and tested on five datasets, and the results look promising.

3. The paper is clearly written and easy to follow.

**Weaknesses:**

The weaknesses of this paper are listed as follows

1. The configuration of the baselines are unclear. It looks like that the paper simply integrate a bunch of off-the-shelf models trained with diferent algorithms as the baselines. However, it is unclear whether these models are trained from the same base model and with a same dataset. Given this, it is hard to conclude whether SEED-GRPO really outperform the baselines

2. In the experiment setup, the maximum output is set to 3000 tokens. However, this might not be enough for hard datasets like AIME24. How would the perforamnce of SEED-GRPO compared to baselines if we allows more output tokens (e.g., 8k)?

3. The argument that prompts inducing high uncertainty harms training lacks an empirical justification. Could the authors conduct a simple experiment, where the model is only trained on those prompts that the model is uncertain and report the performance (we would probably see a performance drop compared to the baseline)?

**Questions:**

See weakness section

---

> ### Author Response · Authors · 2025-11-24
>
> We thank the reviewer **q1DN** for the constructive feedback and for acknowledging that **SEED-GRPO** proposes a simple yet effective method with promising results across extensive benchmarks.  We provide a point-to-point response below.
>
> **Q1**: The configuration of the baselines is unclear. Given this, it is hard to conclude whether SEED-GRPO really outperforms the baselines.
>
> **A1**: We appreciate this important concern. Yes, we control the training configuration for fair comparisons.
>
> We conduct highly controlled comparisons in **Tables 3 and 4a**. Since our codebase is built upon Dr.GRPO [Ref 1], we ensure completely identical settings to demonstrate effectiveness:
>
> - **Same base models:** Qwen2.5-Math-1.5B/7B and DeepSeek-R1-Distill
> - **Same training data:** MATH Level 3-5
> - **Same hyperparameters:** the default setting is maintained exactly as in Dr.GRPO
> - **Same evaluation benchmarks**
>
> We acknowledge that the training configuration comparison in Table 3 of the original draft may not have been sufficiently clear, and **we will revise this** to better highlight the fair comparison settings in the Experiments Section.
>
> Table 1. **Summary of method comparison under identical configuration** (from original draft):
> **Initial Model  is Qwen2.5-Math-7B**
>
> | Method | #Train Data | #Prompt Batch Size | #Rollouts (G) | AIME24 | MATH | AMC | MIN | OLY | Avg |
> | --- | --- | --- | --- | --- | --- | --- | --- | --- | --- |
> | **Initial Model 7B** | – | – | – | – | – | 45.8 | 21.3 | 34.7 | 38.2 |
> | **Dr.GRPO 7B** | 8.5k | 128 | 8 | 43.3 | 80.0 | 62.7 | 30.1 | 41.0 | 51.4 |
> | **SEED-GRPO 7B** | 8.5k | 128 | 8 | 46.7 | 83.0 | 69.9 | 36.7 | 46.8 | 56.6 |
>
> As shown in Table 1, under identical training configurations, SEED-GRPO consistently outperforms Dr.GRPO across all benchmarks, with an average improvement of **+5.2 points**, demonstrating the genuine effectiveness of our proposed method.
>
>
>
>
>
> **Q2**: How would the performance of SEED-GRPO compare to baselines if we allow more output tokens (e.g., 8k)?
>
> **A2**: Thank you for this insightful question. We actually explore this setting in our original submission. **In Table 2 of the draft** (Line 347-348)**, we explicitly demonstrate the performance of SEED-GRPO under an 8k output length setting, which indeed further boosts performance.**
>
> **Key observations:**
>
> - Both models use identical settings except for the output length during training and inference
> - The **SEED-GRPO 7B (Linear, α=0.02, R1-Distill)** is trained and evaluated with 3k maximum output tokens.
> - The **SEED-GRPO 7B (Linear, α=0.02, R1-Distill, 8k length)** is trained and evaluated with 8k maximum output tokens.
>
> We respectfully note that this comparison was already included in our original submission (Table 2 Line 324). For the reviewer's convenience, we directly copy the relevant portion below:
>
> | Method | AIME | AMC | MATH | MIN | OLY | Avg |
> | --- | --- | --- | --- | --- | --- | --- |
> | **SEED-GRPO 7B (Linear, α=0.02, R1-Distill)** | 50.0 | **78.3** | 91.6 | 38.6 | 61.5 | 64.0 |
> | **SEED-GRPO 7B (Linear, α=0.02, R1-Distill, 8k length)** | **63.3** | 74.7 | **93.2** | **40.4** | **65.3** | **67.4** |

---

> ### Author Response · Authors · 2025-11-24
>
> **Q3**: The argument that prompts inducing high uncertainty harms training lacks an empirical justification. Could the authors provide empirical evidence.
>
> **A3**: We appreciate this question and provide empirical evidence supporting our claim.
>
> **Experimental Setup:**
> Instead of simply filtering out low-uncertainty data (which introduces a confounding variable by reducing the total training tokens), we utilized the full dataset but inverted the modulation logic:
>
> - **Standard SEED-GRPO (Ours):** Reduces the advantage (learning signal) for **High-Uncertainty** samples, which is the default setting of our method.
> - **Reverse SEED-GRPO (Counter-factual):** Reduces the advantage for **Low-Uncertainty** samples, thereby forcing the model to prioritize learning from High-Uncertainty samples.
>
> We compare two runs of our standard method (`seed-exp1`, `seed-exp2`) against two runs of the reverse setting (`seed-reverse-exp1`, `seed-reverse-exp2`) across 5 math benchmarks. The results in Table R2 show:
>
> 1. **Consistent Degradation:** The Reverse models consistently underperform the Standard models throughout the training process.
> 2. **Collapse:** Notably, `seed-reverse-exp1` suffers a catastrophic **model collapse at Step 600**, with accuracy plummeting.
>
> **Conclusion:**
>
> These experimental results are consistent with the findings of Ren et al. [Ref 2] and Gekhman et al. [Ref 3], and they empirically justify our core design choice to down-weight high-uncertainty samples.
>
> We thank Reviewer **q1DN** for this insightful question, which motivated us to highlight this important ablation study. These results will be included in the revised submission.
>
> Table R2. Average accuracy (%) across 5 math benchmarks at different training steps.
>
> | Method | Step 112 | Step 192 | Step 224 | Step 560 | Step 608 |
> | --- | --- | --- | --- | --- | --- |
> | seed-exp1 | 48.5 | 50.7 | 49.8 | 52.1 | 52.9 |
> | seed-exp2 | 51.0 | 50.6 | 52.1 | 51.0 | 52.4 |
> | seed-reverse-exp1 | 48.9 | 49.0 | 49.3 | 50.9 | 30.4 |
> | seed-reverse-exp2 | 47.7 | 48.2 | 46.6 | - | - |
>
> While a line plot would better illustrate the training dynamics and the sharp collapse trend, we present the results in tabular form due to rebuttal format constraints.  (We add a line plot to Table 4 of the revised submission.)
>
> **Ref:**
>
> [Ref 1] Liu, Z., Chen, C., Li, W., Qi, P., Pang, T., Du, C., ... & Lin, M. (2025). Understanding r1-zero-like training: A critical perspective. In Conference on Language Modeling (COLM) 2025.
>
> [Ref 2] Mengjie Ren, Boxi Cao, Hongyu Lin, Cao Liu, Xianpei Han, Ke Zeng, Wan Guanglu, Xunliang Cai, and Le Sun. Learning or self-aligning? rethinking instruction fine-tuning. In ACL, 2024.
>
> [Ref 3] Zorik Gekhman, Gal Yona, Roee Aharoni, Matan Eyal, Amir Feder, Roi Reichart, and Jonathan
> Herzig. Does fine-tuning LLMs on new knowledge encourage hallucinations? In EMNLP, 2024.

---

> ### Author Response · Authors · 2025-11-27
>
> Dear reviewer q1DN,
>
> We appreciate your thoughtful review, which has helped us further improve the quality of our work. We have added more
> comprehensive results and figures to improve the readability for readers. Please let us know if you have any other concerns.
>
> Best regards,
> Authors

---

### Official Review · Reviewer_ogpC · 2025-10-30

**Soundness:** 2
**Presentation:** 3
**Contribution:** 2
**Rating:** 4
**Confidence:** 4

**Summary:**

This paper presents SEED-GRPO, an enhancement to Group Relative Policy Optimization (GRPO) algorithm, which incorporates uncertainty-aware prompt reweighting during reinforcement learning. The key novelty lies in introducing semantic entropy, a measure of response diversity across multiple rollouts, as a prompt-level uncertainty signal. Intuitively, the more uncertain the response is, the fewer advantages it should have, and the less magnitude the policy update should be. This enables conservative learning on high-uncertainty prompts and encourages learning on confident cases.

Extensive experiments on five mathematical reasoning benchmarks (AIME24, AMC, MATH, Minerva, and OlympiadBench) show that SEED-GRPO achieves state-of-the-art results, outperforming strong baselines like Dr.GRPO and DisCO even with smaller model sizes (7B vs. 32B).

**Strengths:**

1. Quality: SEED-GRPO achieves state-of-the-art performance on average performance in five mathematical reasoning benchmarks with the Qwen2.5-Math backbone model. Over 15 baselines have been included for comparison.

2. Clarity: The paper is well written and easy to follow.

**Weaknesses:**

1. Significance: The paper focuses exclusively on mathematical reasoning, where uncertainty and correctness are easy to define. It remains unclear how semantic entropy performs in open-ended or multimodal domains, where "semantic clusters" may not be easily defined. This introduces challenges for the algorithm to extend to more general scenarios.

2. Novelty: Although the paper claims it is the first paper to incorporate uncertainty into GRPO, the actual implementation is essentially reweighting prompts based on the final answer's self-consistency, which is not new in GRPO [1]. It would be nice if the authors could tone down this claim.

3. Quality: It would be nice if there were case studies to show that the interpretation in lines 273–300 is true in actual samples.

4. Clarity: The background color of Figure 2 can be improved by using a consistent pure color to improve readability.

### Reference
[1]: Chen, Yi, et al. "GRPO-CARE: Consistency-Aware Reinforcement Learning for Multimodal Reasoning." arXiv preprint arXiv:2506.16141 (2025).

**Questions:**

* It appears the rigorous form of f is not given. May I ask what the specific form of $f$ is?

---

> ### Author Response · Authors · 2025-11-24
>
> We thank reviewer **ogpC** for the valuable time and thoughtful feedback. We provide a point-to-point response below.
>
> **Q1**: The paper focuses exclusively on mathematical reasoning. It remains unclear how the proposed method extends to open-ended or multimodal domains.
>
> **A1**: Thanks for your valuable feedback. We fully agree that extending uncertainty-aware RL beyond mathematical reasoning is an important long-term direction.
>
> **For multimodal domains:**
>
> We extend SEED-GRPO to the vision–language setting by following the setup of Visual-RFT [Ref 1] and GPG [Ref 2] on fine-grained image classification with Qwen2-VL-2B. Concretely, we adopt GPG as the baseline RL algorithm and keep all hyperparameters and data processing **identical**, only replacing the policy optimization with SEED-GRPO. Uncertainty is estimated via semantic entropy, computed by clustering the **final answers** of model rollouts, in the same way as in our math experiments. As reported in Table R1, SEED-GRPO consistently improves over GPG and other baselines on four fine-grained datasets, increasing the average accuracy from 86.0% to 88.5%, which demonstrates that our method is not restricted to purely textual math reasoning but naturally extends to multimodal tasks.
>
> Table R1. 4-shot Results on Fine-grained Classification Datasets.
>
> | Models | Average | Flower102 [Ref 3] | Pets37 [Ref 4] | FGVC [Ref 5] | Cars196 [Ref 6] |
> | --- | --- | --- | --- | --- | --- |
> | Qwen2-VL-2B | 56.0 | 54.8 | 66.4 | 45.9 | 56.8 |
> | + SFT | 55.6 | 58.5 | 55.5 | 67.9 | 40.5 |
> | + GRPO | 81.9 | 71.4 | 86.1 | 74.8 | 95.3 |
> | + GPG | 86.0 | 73.0 | 87.1 | 86.8 | 97.1 |
> | +SEED-GRPO | **88.5** | **78.2** | **89.3** | **88.9** | **97.7** |
>
> **For open-ended domains:**
>
> Original Semantic Entropy papers [Ref 8, Ref 9] utilize NLI models or LLMs (e.g., GPT-4) for semantic clustering, which is naturally designed for open-ended domains such as TriviaQA, SQuAD 1.1, NQ-Open, and SVAMP. While feasible for offline evaluation, this is **prohibitively expensive for online RL training**, which requires computing semantic entropy for thousands of rollout samples at every training step.
>
> We acknowledge that efficient semantic clustering for open-ended domains remains a non-trivial challenge. However, for current domains with **verifiable outcomes** (such as Math, image classification, or other RLVR tasks), our current implementation is highly effective and computationally efficient.
>
> While we appreciate this valuable suggestion, extending our approach to open-ended domains requires substantial research efforts that are beyond the scope of the current rebuttal period. We will explore it in the future work.
>
> In summary, our new multimodal experiments provide concrete evidence that SEED-GRPO transfers beyond math, and we view fully scalable extensions to more open-ended domains as an important direction for future work.

---

> ### Author Response · Authors · 2025-11-24
>
> **Q2**: While the current math-specific implementation relies on clustering final answers, which shares some similarity with self-consistency-based reweighting used in GRPO-CARE [Ref 7]. It would help if the authors could better differentiate the two contributions.
>
> **A2**: We will revise the manuscript to tone down the "first" claim. We will explicitly discuss and cite this work in our revised "Related Work" section. While both methods leverage the intuition of consistency, there are fundamental differences.
>
> 1. Different optimization targets (Advantage Modulation vs. Reward Shaping):
>
> **SEED-GRPO**:  We use semantic entropy — a rigorously derived information-theoretic measure of epistemic uncertainty [Ref 8, Ref 9] — rather than simple majority vote count. We use this uncertainty to create adaptive learning rate for each training sample (prompt).
>
> **GRPO-CARE**: Uses consistency as a **bonus reward signal.**
>
> **2. Implementation**:
>
> - **SEED-GRPO**: **Answer-based clustering is an implementation choice**, not our core contribution. Semantic entropy computation requires clustering, which **can** **be implemented in multiple ways** (NLI models, embeddings models, or simply ChatGPT). We chose answer-based clustering for mathematical reasoning due to its efficiency and effectiveness.
> - **GRPO-CARE**: Focuses on multiple rollouts’ consistency.
>
> We see extending SEED-GRPO to video settings and benchmarking it against GRPO-CARE in that domain as exciting future work. We thank the reviewer **ogpC** for pointing out this promising direction.
> We will:
> 1. Add a detailed comparison with GRPO-CARE in the Related Work section
> 2. Tone down any overclaims about novelty
> 3. More clearly emphasize our specific contribution: **uncertainty-modulated advantage**

---

> ### Author Response · Authors · 2025-11-24
>
> **Q3 :** Request for case studies regarding lines 273-300.
>
> **A3 :** We appreciate this suggestion. To empirically verify the mechanism described in Lines 273–300, we track the **Effective Learning Rate (Effective LR)** for two samples from our training log.
>
> **Parameters:**
>
> - Global learning rate: Lr = 1e-6,
> - α = 0.02,
> - $SE_{max} = \ln(8) \approx 2.079$.
> - f(x) = 1 - α * $(SE(q)/SE_{max})$
>
> **Prompt 1** ='<|im_start|>system\nPlease reason step by step, and put your final answer within \\boxed{}.<|im_end|>\n<|im_start|>user\nReal numbers $a$ and $b$ satisfy the equations $3^a=81^{b+2}$ and $125^b=5^{a-3}$.  What is $ab$?<|im_end|>\n<|im_start|>assistant\n’
>
> Response 1 : "To solve the problem, we need to find the values of \\(a\\) and \\(b\\) that satisfy the given equations:\n\n1. \\(3^a = 81^{b+2}\\)\n2. \\(125^b = 5^{a-3}\\)\n\nFirst, let's rewrite the equations in terms of the same base.\n\n1. \\(3^a = (3^4)^{b+2} \\implies 3^a = 3^{4(b+2)} \\implies a = 4(b+2) \\implies a = 4b + 8\\)\n2. \\(125^b = (5^3)^b \\implies 5^{3b} = 5^{a-3} \\implies 3b = a - 3\\)\n\nNow we have a system of linear equations:\n1. \\(a = 4b + 8\\)\n2. \\(3b = a - 3\\)\n\nWe can substitute the first equation into the second equation to solve for \\(b\\):\n\n\\[3b = (4b + 8) - 3 \\implies 3b = 4b + 5 \\implies b = -5\\]\n\nNow substitute \\(b = -5\\) back into the first equation to solve for \\(a\\):\n\n\\[a = 4(-5) + 8 \\implies a = -20 + 8 \\implies a = -12\\]\n\nSo, we have \\(a = -12\\) and \\(b = -5\\). Now we need to find \\(ab\\):\n\n\\[ab = (-12)(-5) = 60\\]\n\nThe final answer is:\n\n\\(\**\boxed{60}**\\)\n\nLet's verify this solution using Python to ensure accuracy.\n …"
>
> Response 2: “… **\boxed{60}**”
>
> Response 3: “… **\boxed{60}**”
>
> Response 4: “… **\boxed{60}**”
>
> Response 5: “… **\boxed{60}**”
>
> Response 6: “… **\boxed{60}**”
>
> Response 7: “… **\boxed{60}**”
>
> Response 8: “… **\boxed{60}**”
>
> The model is semantically consistent. The modulation factor is 1, allowing the gradient to update the policy with the full learning rate.
>
> **Prompt 2** ='<|im_start|>system\nPlease reason step by step, and put your final answer within \\boxed{}.<|im_end|>\n<|im_start|>user\nSolve the inequality\n\\[\\left| \\frac{2x - 1}{x - 1} \\right| > 2.\\]<|im_end|>\n<|im_start|>assistant\n’
>
>
> Response 1 ="To solve the inequality \\(\\left| \\frac{2x - 1}{x - 1} \\right| > 2\\), we need to consider the definition of absolute value. The inequality \\(\\left| \\frac{2x - 1}{x - 1} \\right| > 2\\) can be broken into two separate inequalities:\n\\[ \\frac{2x - 1}{x - 1} > 2 \\]\nand\n\\[ \\frac{2x - 1}{x - 1} < -2. \\]\ ...”
>
> Response 2: “… **\boxed{\left(-\\infty, \frac{3}{4}\right] \cup [1, \\\infty)}**”
>
> Response 3: “… **\boxed{No answer}**”
>
> Response 4: “… **\boxed{No answer}**”
>
> Response 5: “… **\boxed{\\left(\\frac{3}{4}, 1\\right) \\cup (1, \\infty)}**”
>
> Response 6: “… **\boxed{(-\\infty, \\frac{3}{4}\right] \cup (1, \\infty)}**”
>
> Response 7: “… **\boxed{No answer}**”
>
> Response 8: “… **\boxed{(\\frac{3}{4}, \\infty)}**”
>
> In this example, responses **1 and 5** belong to the same semantic cluster, responses **2 and 6** form another cluster, response **8** forms its own standalone cluster, and responses **3, 4, and 7** each correspond to three separate clusters due to producing “No answer.”
>
> Under these cluster assignments, the semantic entropy is approximately **1.733**, indicating that the model exhibits a noticeable degree of uncertainty for this prompt. According to the uncertainty-aware scaling function defined in the main text (Lines 273–300), the effective learning rate for this question becomes:
>
> $$
> Lr = 1\text{e-}6 \times \left[ 1 - \alpha \cdot \frac{1.733}{2.079} \right] \approx 0.983 \times 10^{-6}
> $$
>
> **Conclusion:**
> This case study confirms: SEED-GRPO dynamically assigns a specific learning pace to each question.

---

> ### Author Response · Authors · 2025-11-24
>
> **Q4** : Improve the readability of Figure2.
>
> **A4** : Thank you for this suggestion. We will improve the readability of Figure 2 in the revised version by changing the background.
>
> **Q5**: Detail forms of f(x).
>
> **A5:** We sincerely thank the reviewer for this important observation. We acknowledge that the detailed mathematical forms of f(x) were omitted in the initial submission for brevity. We will add comprehensive details in the Appendix of the revised manuscript, including the specific functional forms for different domains, architectural details, and implementation specifics. This will ensure full reproducibility of our method.
>
> **Ref:**
>
> [Ref 1] Liu, Z., Sun, Z., Zang, Y., Dong, X., Cao, Y., Duan, H., ... & Wang, J. (2025). Visual-rft: Visual reinforcement fine-tuning. In ICCV 2025.
>
> [Ref 2] Chu, X., Huang, H., Zhang, X., Wei, F., & Wang, Y. (2025). Gpg: A simple and strong reinforcement learning baseline for model reasoning. In NeurIPS 2025.
>
> [Ref 3] Maria-Elena Nilsback and Andrew Zisserman. Automated flower classification over a large
> number of classes. In 2008 Sixth Indian conference on computer vision, graphics & image
> processing, pages 722–729. IEEE, 2008.
>
> [Ref 4] Omkar M Parkhi, Andrea Vedaldi, Andrew Zisserman, and CV Jawahar. Cats and dogs. In CVPR 2012.
>
> [Ref 5] Subhransu Maji, Esa Rahtu, Juho Kannala, Matthew Blaschko, and Andrea Vedaldi. Finegrained visual classification of aircraft. arXiv 2013.
>
> [Ref 6] Jonathan Krause, Michael Stark, Jia Deng, and Li Fei-Fei. 3d object representations for finegrained categorization. In ICCV workshops 2013.
>
> [Ref 7] Chen, Y., Ge, Y., Wang, R., Ge, Y., Cheng, J., Shan, Y., & Liu, X. (2025). GRPO-CARE: Consistency-Aware Reinforcement Learning for Multimodal Reasoning. *arXiv preprint arXiv:2506.16141*.
>
> [Ref 8] Lorenz Kuhn, Yarin Gal, and Sebastian Farquhar. Semantic uncertainty: Linguistic invariances
> for uncertainty estimation in natural language generation. In ICLR, 2023.
>
> [Ref 9] Sebastian Farquhar, Jannik Kossen, Lorenz Kuhn, and Yarin Gal. Detecting hallucinations in
> large language models using semantic entropy. Nature, 630(8017):625–630, 2024.

---

> ### Author Response · Authors · 2025-11-27
>
> Dear reviewer ogpC,
>
> Your insightful comments are greatly appreciated. We provide point-by-point responses to your concerns and are eager to engage in an open dialogue regarding them.  If you have any problems based on our revised submission, we look forward to hearing your valuable feedback.
>
> Thanks.
>
> Authors.

---

### Official Review · Reviewer_UZRX · 2025-10-30

**Soundness:** 3
**Presentation:** 3
**Contribution:** 3
**Rating:** 6
**Confidence:** 3

**Summary:**

This paper introduces SEED-GRPO, an uncertainty-aware extension of Group Relative Policy Optimization for large language models. The key idea is to use semantic entropy to measure the model’s uncertainty about each prompt and adjust the policy update magnitude accordingly. Prompts with consistent responses receive stronger updates, while those with diverse or conflicting answers are updated more conservatively. The method requires no extra sampling cost since it reuses GRPO rollouts for entropy estimation. Experiments on five mathematical reasoning benchmarks show consistent improvements over strong baselines, achieving new state-of-the-art results with a 7B model.

**Strengths:**

1.The idea is clear and intuitive
Using semantic entropy to control update strength is a natural way to make GRPO uncertainty-aware. It seems make sense.

2. The method is simple and clean.
With no extra sampling cost and limited training cost, making it easy to integrate.

3. Experiment results.
Experiments are strong and consistent across five math reasoning benchmarks; improvements over Dr.GRPO and other large baselines are convincing. Ablations are systematic and well presented, showing stable trends across α, weighting functions, and rollout numbers.

**Weaknesses:**

1. Lacks details.
semantic grouping is not clear, will it affect the final performance?

2. The entropy calculation
The semantic entropy is computed only from final answers, will it be better if we also consider the entropy for the thinking process?

3. More benchmark results.
Is it possible to extend the results on more benchmarks, not limited to math?

**Questions:**

Please make response to the points listed in the weaknesses.
Give more details and analysis of the semantic grouping

---

> ### Author Response · Authors · 2025-11-24
>
> We sincerely appreciate reviewer **UZRX**'s thorough review and highly positive assessment of our work. Below, we address each point specifically.
>
> **Q1:** The details of semantic grouping are not clear and will it affect the final performance?
>
> **A1:** Yes, clustering/grouping accuracy critically affects semantic entropy estimation and final performance.
>
> We systematically explored different semantic clustering approaches. Below we detail this iterative process:
>
> 1. NLI-based semantic clustering
>
> In the original semantic entropy work, the authors use language models like RoBERTa-v2, LLaMA, or ChatGPT for clustering [Ref 1, Ref2] via NLI-based pairwise comparison. We initially attempted to follow this approach using RoBERTa-v2 (as in the ICLR version [Ref 1]), loading the 1.5B model on GPU for semantic clustering. However, we quickly discovered a critical incompatibility:
>
> - **The problem**: Mathematical reasoning responses routinely exceed 1,000 tokens. NLI-based clustering requires concatenating response pairs (resulting in 2,000-3,000+ tokens) as input to the model, which far exceeds RoBERTa-v2's maximum input length (512 tokens).
> - **The consequence**: This mismatch produces near-random, inaccurate clustering and unreliable uncertainty estimates. In our early experiments, **SEED-GRPO with RoBERTa-based clustering performed worse than the baseline**, revealing that poor clustering directly degrades performance.
> - **Additional constraint**: RoBERTa-v2 (1.5B parameters) would require significant additional GPU memory during training, while API-based alternatives (ChatGPT, Grok, Gemini, Claude) are prohibitively expensive and slow for online RL.
>
> 2. Naive answer-based clustering fails on "No answer" cases
>
> After abandoning RoBERTa, we pivoted to using the final answer within `\boxed{}` as the clustering criterion—a seemingly natural solution for mathematical problems. However, this revealed another subtle but critical issue:
>
> - **The problem**: Some responses lack a `\boxed{}` answer (especially common in early training stages). Our initial implementation automatically assigned "No answer" as a placeholder.
> - **The bug**: We initially grouped **all "No answer" responses into a single semantic cluster.**
> - **The consequence**: This unreasonable clustering again led to **SEED-GRPO underperforming the baseline** in early experiments.
> 3. Our final solution in the paper
>
> We discovered that the correct approach is to **assign a unique cluster for each “No answer” instance**, preventing merging of semantically distinct rollouts.
>
> This approach is:
>
> - **Computationally efficient**: No additional models or API calls required.
> - **Effective**: Aligns with the mathematical reasoning objective (correct final answer).
> - **Empirically validated**: Enables SEED-GRPO to consistently outperform baselines.
>
> The detail of the Pseudo-code of clustering is provided in **Appendix Part A** (Line 711) of the original draft.

---

> > ### Author Response · Authors · 2025-11-27
> >
> > Dear Reviewer UZRX,
> >
> > We would like to express our sincere gratitude for taking the time to review our work and for your constructive feedback. We are particularly encouraged by your positive comments regarding the simplicity and effectiveness of our approach.
> >
> > We have carefully addressed all of your comments with detailed point-to-point responses. And your feedbacks help us improve the revised paper.
> >
> > As the open discussion phase is ongoing and we have not yet received feedback on our responses, we would greatly appreciate any additional input or clarification you might have based on our revisions.
> > Thank you once again for your valuable time and expertise.
> >
> > Best regards,
> >
> > Authors.

---

> ### Author Response · Authors · 2025-11-24
>
> **Q2**: The semantic entropy is computed only from final answers. Will it be better if we also consider the entropy for the thinking process?
>
> **A2**: We appreciate this insightful suggestion and fully agree that this is a limitation of our current design.
>
> **Our current design is well-motivated**: Focusing on final answer semantic entropy provides an effective and efficient solution, enabling robust clustering and stable uncertainty estimates. This design choice allows us to demonstrate that uncertainty-modulated RL boosts performance in both mathematics  (Table 2 Line 324) and finegrained image classification (see A3 Table R1 below).
>
> While promising, incorporating reasoning process entropy presents **non-trivial methodological challenges** that require careful investigation:
>
> - **Semantic clustering of reasoning paths** is fundamentally more complex than clustering final answers. Unlike final answers (which are often concise and structurally similar), reasoning processes vary significantly in length, structure, and granularity, making it unclear how to define "semantically equivalent" reasoning trajectories. Reasoning paths can differ linguistically but be logically equivalent, or appear similar but diverge in subtle logic.
> - **Inaccurate uncertainty estimation can degrade performance**, even worse than the baseline, as discussed in A1. Since SEED-GRPO directly uses semantic entropy to modulate the advantage, mis-calibrated process-level semantic entropy could systematically misguide policy updates instead of improving them (see new ablation studies in Table 4 of revised submission).
> - **Implementation difficulties in Online RL:** Existing clustering tools are acceptable for **offline** inference, but become impractical in online RL. In the original semantic entropy work [Ref 1, Ref 2], the authors use NLI models, LLaMA, or ChatGPT for pairwise semantic comparisons. This is reasonable when computing entropy offline for evaluation or analysis. However, integrating these heavy computational modules into the **inner loop of Online RL** is impractical. In an online setting, the reward signal must be computed for every generated sample in every training batch.
> Invoking auxiliary NLI models or large LLMs for pairwise comparisons of intermediate reasoning steps would introduce massive computational overhead, drastically reducing training throughput (samples per second). Unlike offline evaluation, which happens once, online reward calculation happens continuously, making the cost of such dense verification prohibitive and destabilizing to the training pipeline.
>
> Given these challenges, naively adding process-level entropy would go beyond the scope of this submission. We sincerely thank the reviewer again for pointing out this promising direction. We have explicitly included this in our Future Work section and will make it a key priority for our future research.

---

> ### Author Response · Authors · 2025-11-24
>
> **Q3**: Is it possible to extend the results to more benchmarks, not limited to math?
>
> **A3**: We appreciate this suggestion.
> We extend our experiment on VLM per your request. We conduct 4-shot experiments on fine-grained image classification with Qwen2-VL-2B following Visual-RFT [Ref 8] and GPG [Ref 9]. We adopt GPG as our baseline RL algorithm and keep all hyperparameters identical, only integrating the advantage modulation with SEED-GRPO. Uncertainty is estimated via semantic entropy by clustering the final answers of model rollouts, exactly as in our math experiments.
>
> Table R1. 4-shot Results on Fine-grained Classification Datasets.
>
> | Models | Average | Flower102 [10] | Pets37 [11] | FGVC [12] | Cars196 [13] |
> | --- | --- | --- | --- | --- | --- |
> | Qwen2-VL-2B | 56.0 | 54.8 | 66.4 | 45.9 | 56.8 |
> | + SFT | 55.6 | 58.5 | 55.5 | 67.9 | 40.5 |
> | + GRPO | 81.9 | 71.4 | 86.1 | 74.8 | 95.3 |
> | + GPG | 86.0 | 73.0 | 87.1 | 86.8 | 97.1 |
> | +SEED-GRPO | **88.5** | **78.2** | **89.3** | **88.9** | **97.7** |
>
> Table R1 reports the results on four standard fine-grained classification datasets. SEED-GRPO consistently improves over GPG and all other baselines on both the average score and on each individual dataset, increasing the average accuracy from 86.0% to 88.5%. Importantly, these results were achieved without extensive hyperparameter tuning, suggesting promising room for further improvement. We will add the results in the revision.
>
> We sincerely thank Reviewer **UZRX** for this suggestion. A more systematic exploration of a broader range of tasks is an important and valuable direction for future work.
>
> **Ref:**
>
> [Ref 1] Lorenz Kuhn, Yarin Gal, and Sebastian Farquhar. Semantic uncertainty: Linguistic invariances
> for uncertainty estimation in natural language generation. In ICLR, 2023.
>
> [Ref 2] Sebastian Farquhar, Jannik Kossen, Lorenz Kuhn, and Yarin Gal. Detecting hallucinations in
> large language models using semantic entropy. Nature, 630(8017):625–630, 2024.
>
> [Ref 3] Liu, Z., Chen, C., Li, W., Qi, P., Pang, T., Du, C., ... & Lin, M. (2025). Understanding r1-zero-like training: A critical perspective. In Conference on Language Modeling (COLM) 2025.
>
> [Ref 4] Li, G., Lin, M., Galanti, T., Tu, Z., & Yang, T. (2025). DisCO: Reinforcing Large Reasoning Models with Discriminative Constrained Optimization. In NeurIPS 2025.
>
> [Ref 5] Jixiao Zhang and Chunsheng Zuo. 2025. GRPO-LEAD: A Difficulty-Aware Reinforcement Learning Approach for Concise Mathematical Reasoning in Language Models. In EMNLP 2025.
>
> [Ref 6] Zhang, Q., Wu, H., Zhang, C., Zhao, P., & Bian, Y. (2025). Right question is already half the answer: Fully unsupervised llm reasoning incentivization. In NeurIPS 2025.
>
> [Ref 7] Shenzhi Wang, Le Yu, Chang Gao, Chujie Zheng, Shixuan Liu, Rui Lu, Kai Dang, Xionghui
> Chen, Jianxin Yang, Zhenru Zhang, et al. Beyond the 80/20 rule: High-entropy minority tokens
> drive effective reinforcement learning for llm reasoning. In NeurIPS 2025.
>
> [Ref 8] Liu, Z., Sun, Z., Zang, Y., Dong, X., Cao, Y., Duan, H., ... & Wang, J. (2025). Visual-rft: Visual reinforcement fine-tuning. In ICCV 2025.
>
> [Ref 9] Chu, X., Huang, H., Zhang, X., Wei, F., & Wang, Y. (2025). Gpg: A simple and strong reinforcement learning baseline for model reasoning. In NeurIPS 2025.
> [Ref 10] Maria-Elena Nilsback and Andrew Zisserman. Automated flower classification over a large
> number of classes. In 2008 Sixth Indian conference on computer vision, graphics & image
> processing, pages 722–729. IEEE, 2008.
>
> [Ref 11] Omkar M Parkhi, Andrea Vedaldi, Andrew Zisserman, and CV Jawahar. Cats and dogs. In CVPR 2012.
>
> [Ref 12] Subhransu Maji, Esa Rahtu, Juho Kannala, Matthew Blaschko, and Andrea Vedaldi. Finegrained visual classification of aircraft. arXiv 2013.
>
> [Ref 13] Jonathan Krause, Michael Stark, Jia Deng, and Li Fei-Fei. 3d object representations for finegrained categorization. In ICCV workshops 2013.

---

### Meta-Review · Area_Chair_t3H2 · 2026-01-08

**Summary:**

This paper proposes SEED-GRPO, which integrates semantic entropy into GRPO to enable uncertainty-aware policy updates for LLM training. The core idea is to measure model uncertainty via semantic entropy computed from multiple rollout responses, then use that to reweight the advantage function.

During the review period, several concerns were raised by the reviewers, particularly regarding novelty in relation to GRPO-CARE, which was a point of concern shared by most reviewers. Additionally, the original motivation of this paper is to leverage semantic information rather than merely the final answer to address issues such as hallucinations and degraded downstream performance. However, in the actual experiments, the "semantic entropy" appears to be calculated using only the final answer. This makes it unclear to me whether the experiments actually validate the originally proposed idea, or whether the implemented algorithm should be categorized as a baseline that only uses final answers. Given these concerns, I do not believe this paper is ready for publication at ICLR.

**Reviewer Concerns:**

I think concerns about experimental details or additional experiments are (partially) addressed, but major concerns like novelty or only using final answer to calculate "semantic entropy" are still outstanding.

**Reviewer Scores:**

It is likely that all reviewers will remain their scores.

---

### Decision · Program_Chairs · 2026-01-26

Reject